# Unfreezing the Discursive Hegemonies Underpinning Current Versions of "Social Sustainability" in ECE Policies in Anglo–Celtic, Nordic and Continental Contexts

Alicja R. Sadownik [1,*], Yvonne Bakken [1], Josephine Gabi [2], Adrijana Višnjić-Jevtić [3] and Jennifer Koutoulas [4]

1   Faculty of Education, Arts and Sports, Western Norway University of Applied Sciences, 5063 Bergen, Norway; Yvonne.Bakken@hvl.no
2   Education and Social Research Institute, Manchester Metropolitan University, Manchester M15 6GX, UK; j.gabi@mmu.ac.uk
3   Faculty of Teacher Education, University of Zagreb, 10000 Zagreb, Croatia; adrijana.vjevtic@ufzg.hr
4   Early Years Intercultural Association, Liverpool, NSW 2170, Australia; jkoutoulas@eyia.org.au
*   Correspondence: Alicja.Renata.Sadownik@hvl.no

**Abstract:** Social sustainability is linked to finding new ways of living together and strengthening social capital and participation, as well as to social justice and equity in societies, and it is becoming increasingly important for diverse multicultural societies. In this article, we trace understandings of social sustainability as established in Early Childhood Education (ECE) policy documents by following the chains of meaning connected to sense of belonging, local place and cultural diversity and through ECE collaboration with children's parents/caregivers. Critical discourse analysis has been applied to trace the chains of meaning attached to these concepts in ECE steering documents in Australia, Croatia, Denmark, Norway, Poland, Serbia, Slovenia, Sweden and the UK (England, Scotland, Wales and Northern Ireland). Such analysis shows different ways in which the ECE polices indirectly work with social sustainability, as well as create critical distance from the sets of meanings established in each country (by proving a chain of meaning established in the policy documents of another country). In conclusion, we do not advocate in favour of any of the chains of meaning but argue for continual reflection and reflexivity, and we see research to be a particularly significant arena in which to unfreeze the taken for granted and sustainable notion.

**Keywords:** social sustainability; belonging; collaboration with caregivers; place and space; cultural diversity

## 1. Introduction

Among researchers of early childhood education for sustainability, there appears to be joint agreement on the necessity of balancing the discursive domination of the environmental pillar and generating knowledge and reflection connected to social and economic sustainability [1–10]. Social sustainability that embraces good, equity-based and new ways of living together is not far from ECE policies and practices. In this paper, we ask how social sustainability is more or less directly written into the ECE curricula of 12 countries.

In order to answer the question posed in this article about social sustainability in ECE curricula, we begin with a short description of our study's methodology, followed by an analysis of the concepts that we have seen as operationalising social sustainability at the level of ECE curricula. The concepts of belonging, diversity, local place and collaboration with parents/caregivers are firstly described using diverse theories, followed by a study of their presentation in the analysed policy documents. In the discussion section, we try to show how meanings occurring in one policy can visualise what is excluded in another or how a set of meanings established in theories show what is excluded from policy discourses.

This process reconstructs the foundations of the discursive hegemonies that shape the social ECE policies that indirectly design ECE work with social sustainability.

> *How and why do we operationalise social sustainability in terms of belonging, diversity, local place and collaboration with parents/caregivers?*

According to Eizenberg and Jabareen [11], social sustainability refers to the concepts of equity and social justice, which allow all members of a society, regardless of diverse categories of differences, to participate in a community as equal citizens. Hägglund and Johansson [6] operationalise these aspects of social sustainability in the context of ECE as belonging. Children's sense of belonging to their peer group in the institutional setting of ECE is recognised by Hägglund and Johansson as a wide and sensitive concept that embraces the daily dynamics of being included/excluded, of participating or not participating in diverse peer communities. Sense of belonging embraces the negotiations over a child's position in play as well as being part of the peer community, in general. Research on the sense of belonging, however, also identifies those who do not belong, who do not have access to membership in a particular group [12]. Such research, by reconstructing diverse categories that "do not belong", connects to categories of difference and to diversity [13]. This is why, in our opinion, the concept of diversity, as an endless possibility of being different from those who belong (as well as being different among those who belong), should be included in discussions of belonging and, thus, of social sustainability.

Sense of belonging does not relate solely to people; it also relates to place and locality. A strong sense of "belonging" to a place, either consciously or through everyday behaviour, such as participating in place-related affairs, would be indicative of a "sense of place" [14] (p. 24), which is why local places can be seen as relevant to social sustainability [15,16]. Contextualisation of ECE in local communities is factualised when a child enters an ECE setting, firstly by and through their parents and caregivers. The links between departure from individual sense of belonging and embracing diversity, local place and community, and parents and caregivers will be included in our analysis, as these are relevant to social sustainability.

Our understanding of ECE-related social sustainability thus departs from children's communities and includes work with diversity within the ECE setting, (diverse) families and parents and the place and community that constitute the local ECE context. Even though these issues are not always directly linked by the diverse national curricula to social sustainability, the UNESCO report, "The contribution of early childhood to a sustainable society" [17], points out the role that ECE plays, nevertheless, in developing values, behaviours and skills that have a great impact on furthering socially sustainable attitudes and actions. Moreover, EU policy documents [18–20] formulate ECE sector goals, such as social cohesion, social inclusion, poverty reduction and migration integration, which relate the sector's daily work to social sustainability, even without articulating a direct link. Therefore, we have decided to trace the indirect social sustainability policies expressed in the ECE curricula of the 12 represented countries. On the basis of the UNESCO report [17], we have assumed that ECE policies of belonging, diversity, local place and collaboration with parents/caregivers are policies for social sustainability. In other words, issues of social sustainability are addressed in the guidelines for ECE work with children, both when building a sense of belonging and in their relations to the outside community.

We have identified a large number of ECE curricula around the world which, even if they do not directly refer to social sustainability, do refer to children's sense of belonging or their inclusion, collaboration with caregivers and local place/region. This is why we have chosen to reconstruct social sustainability in ECE policies by tracing chains of meaning attached to sense of belonging, diversity, local place and collaboration with parents/caregivers. We aim to reveal the sets of meanings that underpin the social sustainability policies in Australia, Croatia, Denmark, Norway, Poland, Serbia, Slovenia, Sweden and the United Kingdom (England, Wales, Scotland and Northern Ireland) by undertaking a comparative analysis.

It is important to emphasise that our analysis considers policies and not institutional practice in ECE settings. Our conclusions may subsequently relate to the practice of policymaking, which means that the reader will not be directly encouraged to make improvements in daily practice within an ECE setting.

## 2. Methodology: Critical Inquiry Tracing Chains of Meanings

The research questions driving our analysis address the chains of meaning attached to the four chosen concepts in policy documents in 12 countries. The methodology to be applied thus needed to provide us with a theoretical toolkit that allows such an analysis. Laclau and Mouffe [21], when explaining the establishment of meaning, indicate the relationship between the signifier, the sign and the signified, where the signifier is the word or sound designating a particular object (the sign) as a mental concept (the signified). Eventual negotiation, variation or change in meaning, in this sense, relates to the possibility of a different relating of a particular signified. In our analysis, we will focus on the signifiers (words) and the signifieds (concepts) in terms of policy analysis, and, for this reason, we have excluded physical objects.

According to Laclau and Mouffe [21], what stabilises and "freezes" a particular relationship between a signifier and a signified is discourse, and what also happens in this process is the exclusion of other possible meanings (signifieds). If we take "child" as a signifier, we can relate it to a signified, such as "adult dependent" or "citizen", each of which will establish another totality of meaning. Each of these will be based on the exclusion of all other possible signifieds connected to "child" [22]. The excluded signifieds, or the signifieds that are excluded from the created meaning, create a reservoir of possible meanings called the "*surplus of meaning*" or the "*field of discursivity*" [21] (p. 111). It is the "excluded rest" that, according to Laclau and Mouffe [21], will always try to enter and challenge the dominant discourse, the established meaning. In our analysis, the "rest" that is excluded from the discourse of a given country's ECE policy may appear in that of another country and, in this way challenge the dominant set of meanings within the analysed administrative entity.

The discourse in which the child is an "adult dependent" presents the excluded surplus of meaning from the discourse in which the child is a "citizen", yet neither of these may include violence against children as their signified. If we take as our point of departure the issue of violence against children, these two discourses regarding the child who is an adult dependent and a competent citizen will be woven into a chain of equivalence, which will make the difference between them much less visible. The chain of equivalence between sets of meanings that do not initially belong together starts by relating them to a common project/goal as well as by defining the forces to be opposed, the "enemy" [23] (p. 50). This implies that the meanings in the discourses (even where initially very different) become equivalent when fighting against a common enemy.

The four concepts chosen for our analysis are seen as equivalent in relation to social sustainability, and, as such, they are different from, or the opposite of, the environmental or economic aspects of sustainability. Although the dimensions of sustainability are considered to overlap in many respects [24], we will treat them as opposite entities in this article, as it is the dominance of environmental aspects of sustainability [8] that have made us explore its other, non-environmental, aspects (i.e., social sustainability). Having established this opposition, we developed a chain of concepts that we saw as equivalent in relation to social sustainability, as presented by Hägglund and Johansson [6] and in relation to the language of ECE.

As noted in the introduction, the four concepts we decided to trace occur in the ECE steering documents in the following countries, although social sustainability, itself, does not necessarily appear in them. This is why we intend to trace the existing social sustainability-related meanings that frame ECE work with social sustainability.

The concepts of belonging, diversity, local place and collaboration with parents/caregivers have been traced in the indicated ECE steering documents of the following countries:

**Australia**: "Belonging, Being and Becoming. Early Years Learning Framework for Australia" [25].
**Croatia**: "Nacionalni kurikulum za rani i predškolski odgoj i obrazovanje" [26].
**Denmark**: "The strengthened pedagogical curriculum." Framework and content [27].
**England**: "Statutory Framework for the Early Years Foundation Stage: Setting the Standards for Learning, Development and Care for Children From Birth to Five" [28].
**Birth to Five Matters**: "Guidance for the Sector by the Sector" (in consultation phase) [29].
**Northern Ireland**: Curricular Guidance for Pre-School Education. Belfast: Council for the Curriculum, Examinations and Assessment [30]."
**Norway**: "Framework Plan for Kindergarten: content and tasks" [31].
**Poland**: "Podstawa programowa wychowania przedszkolnego i kształcenia ogólnego dla szkoły podstawowej. Wychowanie przedszkolne i edukacja wczesnoszkolna" [32].
**Scotland**: "The Early Years Framework." Edinburgh [33].
**Serbia**: "Pravilnik o opštim osnovama predškolskog programa" [34].
**Slovenia**: "Kurikulum za vrtce" [35].
**Sweden**: "Curriculum for the preschool, Lpfö 18" [36].
**Wales**: "Curriculum for Wales: Foundation Phase Framework. Cardiff: Department for Education and Skills" [37].

Separate analytical tables were created for each country using a collaborative file-hosting service (Google Docs). Each table contained quotes relating to the chosen concepts: sense of belonging, diversity, local place and collaboration with caregivers. Our joint but synthetic interpretation of these quotes was put in another column. During three online meetings (of two hours each), we traced diverse chains of equivalence and differences between meanings connected to these concepts in each of the analysed policy documents. The policy documents from the different countries were analysed as part of the wider legal, societal and cultural contexts that each country/entity represents.

The countries chosen for our analysis furthermore represent very different ECE approaches and traditions [38]: the Anglo–Celtic [25], the Nordic [39] and the Continental (post-communist).

## 3. Analysis: The Traced Chains of Meaning

We began with the sense of belonging, which, according to Hägglund and Johansson [6], directly points to the operationalisation of social sustainability in the ECE sector. As the understanding of sense of belonging within the steering documents in some cases embraces and/or relates to diversity and difference, local place and community, as well as to family, we can say that, not only was it theoretically justifiable to include these in our analysis, but that they have also appeared in the research material (policy documents).

Each of the concepts is introduced together with theoretical mapping and followed by analysis of the policy documents.

The curricula from the various countries are not equally represented in the descriptions below, as this depends on the topic-related content in the documents. Therefore, we start the analysis by offering the reader a very synthetic overview of the chosen concepts and understandings extracted from the documents presented in Table 1. The Table 1 can thus serve as a general platform and a simple overview, and further in the article, we will deepen this and present diverse nuances. We would, however, emphasise that the summaries are based on our interpretation of what we see as the core issue to emphasise and relate to. The Croatian, Polish, Slovenian and Polish curricula, which were not available in the English language, were translated for the author team by researchers from the team who had cultural and linguistic access to these countries. The collective work on the summaries was thus based on the unofficial translations delivered by particular individuals.

Table 1. Overview over extracted meanings connected to analysed concepts in all countries' curricula.

| | Sense of Belonging | Local Place and Community | Cultural Diversity | Cooperation with Parents/Caregivers |
|---|---|---|---|---|
| Australia | Belonging is experienced by the child through interconnectedness with others to build a sense of identity. | Children are seen as explorers and learn with others in the local and wider community to develop appreciation for different ways of knowing. | Children's identity is derived from their culture, and they have the right to maintain it. Educators respect cultural diversity, support cultural competence and honour differences. | Partnerships with families are one of the five principles that underpin children's learning outcomes. Reciprocal partnerships are integral to understanding expectation, deepening knowledge and working together professionally. |
| Croatia | Sense of acceptance and belonging are prerequisites for children's social wellbeing. | Kindergarten should establish a partnership with the wider social community, and the child is an active citizen who participates in shaping community. | Children should understand and accept others and their differences in an inclusive environment. | Partnership with families is one of the main principles of the curriculum, and parents are involved in institutional governance. |
| Denmark | Sense of belonging is related to the process of (minority) integration and becoming part of Danish society, as well as to developing social cohesion. | The pedagogical curriculum should state how the ECE setting involves the local community (in terms of nature and culture) in establishing the holistic learning environment for children. | The pedagogical offer of the ECE setting should be relevant for all children, regardless of their background, language, culture or traditions. | ECE staff should cooperate with parents in relation to both the individual child and the community of children in the ECE setting |
| England | Sense of belonging is not specifically mentioned in the Early Years Foundation Stage curriculum; emphasis is on equality of opportunity, antidiscriminatory practice and ensuring that every child is included and supported. | Settings are required to provide guidance for children to make sense of their physical world and their community through opportunities to explore, observe and find out about people, places, technology and the environment. | This is not mentioned in the ECE curricula, but ECE is obliged to follow the Equality Act 2010 (which explains the provisions for reasonable adjustments). | Emphasis is on a strong partnership between practitioners and parents/caregivers in order to support children's learning at home and in ECE. |
| Northern Ireland | Children develop a sense of belonging through becoming familiar with daily routines in the ECE setting. | Children develop an understanding of space in order to consider the relationships between (human and non-human) objects. | Children should be supported in recognising and valuing the diversity that other children bring to the setting. | Partnership with parents/guardians/carers is at the core of practice and sustaining positive home learning environments. |
| Norway | Sense of belonging is described as coming about through (inclusive) relationships within the peer group and sense of community among children. | Local place is understood as the possibility of using the ECE surroundings during pedagogical work, as well as places that children may be familiar with. | All children are to experience ECE as a place for them. Children are to be introduced to diverse ways of living, thinking and acting, without making any child the representative of any culture/nation/religion. | ECE is to work in understanding and collaboration with children's homes in order to safeguard all-side development. The children should not experience conflicts of loyalty between home and ECE, and, in case of any value-related conflict, the parents need to respect the values of the ECE curricula. |

| | **Sense of Belonging** | **Local Place and Community** | **Cultural Diversity** | **Cooperation with Parents/Caregivers** |
|---|---|---|---|---|
| Poland | This is mentioned in relation to the peer group. | Children are to become familiar with local places and their institutions. The curriculum seems to assume that the localities are urban. | This is not mentioned (apart from national minorities, such as Kashubian). | At the individual level, the parents are receivers of information about the child's developmental progress. At the collective level, the parents can influence the pedagogy and economy of the ECE setting. |
| Scotland | Settings should provide induction activities that help children to settle quickly and to have a sense of belonging. | Communities are enabled to develop their own aspirations and outcomes. | Children should learn about their own and other cultures as a way of promoting diversity. | Parents are supported by providing the children with a stimulating learning environment (as realisation of social solidarity). |
| Serbia | The child is meant to acquire a sense of belonging and master how to function in social groups. | Working and partnering with the local community are regarded as necessary for living with the locality (and its local crafts). | The aim of ECE is to develop relationships and gain experience and knowledge of other people. Minorities are recognised as valuable members of society. | The partnership between experts and caregivers is seen as a key element; in the case of dysfunctional families, ECE institutions are seen as supplementary to family care. |
| Slovenia | Everyday life in kindergarten (daily routines, rituals, events, agendas etc.) must give a child a sense of belonging. | One principle of the curriculum is cooperation with the environment as a natural and socio–cultural learning resource. | The aim of the curriculum is the creation of conditions for greater expression and awareness of group differences. | Partnership is expressed by way of parents' rights in relation to institutions, but parents are recognised as valuable partners in education. |
| Sweden | The work team should show respect for the individual and help to create a democratic climate in the preschool, where children have the opportunity to feel a sense of belonging and to develop responsibility and solidarity. | The work team should create the conditions for children to become familiar with their surroundings and those societal functions that are important for everyday life and to take part in local cultural life. | The preschool should provide each child with the conditions to develop their cultural identity and knowledge of and interest in different cultures and an understanding of the value of living in a society characterised by diversity, as well as an interest in local culture. | The preschool should cooperate in a close and trusting fashion with the home, ( . . . ) maintain ongoing dialogue with the child's guardians about the child's wellbeing, development and learning and conduct dialogue about the child's development. |
| Wales | Sense of belonging is defined in relation to children's understanding of Welsh heritage, literature, arts and religious background, as well as the Welsh language. | Children should learn to demonstrate care, responsibility, concern and respect for all living things and the environment. | Children should have an understanding of their own Welsh identity and treat people from all cultural backgrounds in a respectful and tolerant manner. | ECE settings are required to involve parents in daily pedagogical practice to ensure the continuity of children's learning. |

### 3.1. Sense of Belonging

3.1.1. Theoretical Mapping

Sense of belonging is connected to membership in a particular group or entity, which implies that the group is not for all but for "us". This again introduces the struggle of who belongs, who is excluded and who gets to decide [40]. Sumsion and Wong [13], when dealing with these questions, point to three axes of belonging: (1) categorisation, (2) resistance and desire and (3) performativity. Categorisation is related to the core of power relations [41] that underpins the criteria for membership. These may either be related

to externally established categories of difference, such as gender, social class, ethnicity or age, or developed internally by the group [42]. Boldermo's [42] research shows how children's moments of togetherness not only go beyond the socio–politically established categories of difference but also continually change. This can be related to the second aspect of belonging, resistance and desire, which triggers the individual to be hostile to the "given" distinctions and develop a new order of membership. The third aspect of belonging, performativity, embraces the continual negotiations of one's own membership and position in one or another group [16].

Research on children's communities of play has registered the continual negotiation of one's own position in the group [43,44] and the right to undertake a particular role/task/activity [45]. This is in line with Gabi's [46] rhizomatic, fluid and dynamic understanding of belonging. According to Öhman [44], the grouping processes can be facilitated, by which she means the criteria for membership can be extended so that all children can experience belonging. Extending the criteria for membership is also in line with Brown's [47] portrayal of a sense of belonging as being part of a group because of who you are and not because you are fitting in (which, again, is related to being accepted for being like everyone else).

There is a large body of research discussing the sense of belonging that assumes belonging is a fulfilled need for relatedness [48,49] and that focuses on how it influences an individual's other activities. In such research, sense of belonging is reported as having a direct influence on children's motivation and their dedication to activities, as well as the confidence with which they participate in various tasks or activities [50]. This school of thought maintains that a sense of belonging is directly connected to children's wellbeing, with children feeling they are part of a greater system/environment and being more enthusiastic, happier, more interested and more confident [50]. This understanding of belonging implies pedagogical work that facilitates the fulfilment of the need for relatedness. This, however, can be about extending the criteria for the child's membership in a group or about teaching the child how to fit in or presenting to the child where he or she belongs.

### 3.1.2. Policy Analysis

All the policy documents frame the work of ECE services; their understanding of belonging assumes that the respective ECE service is capable of facilitating it in one way or another. There appear to be diverse chains of different meanings and assumptions that are attached to belonging. The main reconstructed difference is related to belonging assumed to be a "fixed" and "fixable", or "performative" and "processual", phenomenon. Our analysis shows that the understandings of belonging as fixed or fixable can, in some policy documents, develop a chain of equivalence in which belonging is understood as fitting in and obeying the social norms, as in the case of the Serbian curriculum, or where belonging is understood as a child's social skill that manifests itself in their being able to feel and explain their own relationships to diverse social groups, as in the case of the Polish [32] curriculum. Such a hegemony of meanings excludes the discourse on belonging established in the Norwegian [31] and Danish [27], as well as the English [28], Welsh [37] and Australian [25], policy documents, which depart from processual and performative understandings of belonging and connect it to the practitioner's work. The practitioner's work should, then, focus on extending the criteria for membership by fostering an appreciation of diversity in children's groups, which in the Danish [27] and Norwegian framework plans for ECE [31] is connected to democratic values.

The Danish Framework Plan [27] locates belonging in the children's community, participation in which seems to be a "natural" outcome of being a part of it. This is, again, related to the experience of democracy: By participating in communities with others, children gain a basic experience of belonging to such communities, as well as an understanding of democracy and democratic processes [27] (p. 36). The experience of belonging is here in a dialectic process with participation in the community, as it both facilitates participation and is strengthened by it. The Norwegian curriculum [31] also underlines the importance of

the children's community and everyone's participation in it and recognises it as connected to democracy. However, it is not connected to a sense of belonging. In the Norwegian curriculum, belonging is expressed in one line along with other values that ECE is to build on: "Meeting every child's need for care, security, belongingness and respect and enabling the children to participate in and contribute to the community are important values that shall be reflected in kindergarten" [31] (p. 7). The Australian curriculum identifies sense of belonging in a way similar to the Norwegian curriculum, as it describes belonging as children's bond with their family and their connection with others. It is the relationships and sense of belonging with them that "shape who children are and who they become" [25] (p. 7). The Danish way of articulating belonging takes its departure from children's activities and participation as phenomena that make sense of belonging occur, which enables the practitioner's work in facilitating diverse ways of participation that are relevant to children's interests, age, abilities, etc.

In contrast, the understanding of belonging in the Swedish curriculum [36] is related to children's more or less fixed linguistic and cultural identities in respect to diverse national minorities operating outside the ECE setting. "Children belonging to national minorities, which include the indigenous Sami people, should also be supported in their language development in their national minority language and encouraged in their development of a cultural identity" [36] (p. 9). The English [28,29], Welsh [37] and Australian [25] understandings of belonging seem to be equivalent to this, as they also relate belonging to children's cultural identities established outside of ECE settings. In this case, the ECE setting becomes an arena where this belonging is played out, and with the help of inclusive practices (the staff's work), it encourages opportunities for the children to "develop a positive self-image and a sense of belonging as part of different communities and have an understanding of their own Welsh identity" [37] (p. 10). The efforts here are not focused on making all the children fit an ideal of Welshness but on extending being Welsh in a way that combines the children's ECE-based experience with their home cultures. The English [29] understanding seems to be equivalent to the Welsh one in that "developing sense of belonging is an important part of inclusive practice" [29] (p. 15). Here [29], however, the children and families belonging to a wider community are seen as primary and fixed, and ECE becomes "only" an arena for promoting and celebrating (not forming) these identities: "Feeling different or being marginalized can lead to feelings of social isolation. When children and their families are able to develop a sense of belonging to a wider community this can reduce these feelings and provide children with a more secure base, from which they can learn, develop and flourish" [29] (p. 15). The children's trajectory for flourishing seems to follow the cultural line of their home cultures, and a different way of forming their identity is not discussed in the policy document (even though it is possible in practice).

This excluded possibility of identity formation/becoming comes up in the Australian curriculum, however, where the focus on exploration and becoming is more explicit and does not define the child (exclusively) through his or her home culture. Moreover, it encourages the child to "explore the diversity of culture, heritage, background and tradition" [25] (p. 30). The Australian framework [25], however, is also equivalent to the English [29] one in that it attributes a strong sense of identity to the children [25] (p. 26), which is to be recognised and performed in the ECE setting (which again may be interpreted as limiting the exploration).

Nevertheless, the general goal of strengthening sense of belonging is the child's general wellbeing. The focus on wellbeing is made explicit in the Croatian curriculum [26]. However, the Croatian curriculum combines the sense of belonging with the sense of being accepted by the group [26]. This implies the possibility that belonging is understood as "fitting in" and adjusting to the group (rather than being included when being oneself). Although it is not clearly stated in the document what it means to be accepted by the group, it is possible to interpret sense of belonging as "fitting in".

The set of meanings related to "fitting in" are more explicit in the Serbian curriculum, which describes sense of belonging as occurring when one acts in line with general social norms and the basic rules for functioning in a group [34]. This implies that belonging to a group is achieved by obeying its rules and norms, which again results in a feeling of being part of the group. The Slovenian framework [35], despite its geographical and cultural proximity to the Serbian one, seems to break out of this chain of meaning by relating sense of belonging to the staff's work and pedagogical efforts (which result in every child having the experience of being part of the group). All activities, daily routines, events and agendas for each day of the week are planned with the intention of giving the children a sense of belonging. The focus of the staff's work on the inclusive character of all activities also receives strong emphasis in the Norwegian, Danish, Swedish, English, Welsh, Scottish and Irish documents.

In Poland, however, the practitioner's work is to focus on preparing the child for school, as the curriculum states that the child who is ready for school is able to "feel and explain his or her own belonging to his or her family, nation, peer group, gender group or other group, for example theatre or sport group" [32] (p. 7). The Polish understanding not only treats belonging to diverse groups as fixed, but it also relates it to the child's ability to feel and explain, which starts one more chain of meanings and possible pedagogical practices that facilitate the child's ability to feel and explain.

### 3.2. Diversity and Difference (and Becoming in the Context of Diversity)

#### 3.2.1. Theoretical Mapping

Works in which belonging is connected to extending the criteria for membership address dealing with difference and diversity. This is related to the *categorisation* aspect of a sense of belonging and also to the *resistance* to overcoming divisions and distinctions and the *desire* to do so [13]. Siraj-Blatchford, Smith and Samuelsson [51] refer to this as an ethos of compassion and respect for difference, equality and fairness, so that inclusive educational experiences can be fostered for all children.

Again, the categories of difference are often related to identity in the sense of it being essentialised and fixed, with the result that individuals are locked into belonging to some but not other social groups [13]. While some researchers demonstrate that making a child an ambassador and representative of the family culture may be ethically problematic [52], others argue for making all content brought by the children to ECE settings equally valued, regardless of cultural, religious, linguistic or historical background [46].

Attributing a particular cultural identity to a child or allowing the child to self-create their own sense of self in dialogical engagement with the diverse cultural values and meanings available in a diverse society is a question of *becoming*. *Becoming*, which can also be put as *bildung* or *cultural formation* [53], is described as taking place through dialogical involvement with the diverse cultural values and meanings that exist in the community and in interactions with other individuals/generations, as well as in artefacts [53,54]. It is thus a social and mutual process through which, in the ECE institutional context, "children and teachers shape themselves and are shaped in dialogical processes with other people, culture and history, nature and society" [55] (p. 50), and diversity in the group (of family and/or children) can function as a great resource. The question, however, is the degree to which particular children are to be representatives/ambassadors of their family cultures, and the degree to which ECE is to present diverse cultural values and meanings as a context for everyone's formation, which could allow the children to decide on the content with which they identify and when, as well as the content with which they do not identify (without making them responsible for representing one culture or nation or another). Making diversity the context of everyone's becoming opens the door to "unlimited possibilities for 'becoming' across accessible cultural values, meanings and heritage in the intercultural context of ECE" [52].

One can say that in such an intercultural context, the processes of becoming intensify and grow more complex and immersive [53,54]. This requires reflection on the part of

practitioners with regard to how diverse cultures are recognised in ECE settings and whether these settings function such that certain cultures are represented by particular children or whether they provide a context for everyone's becoming [52]. Becoming, as a process, starts with explorative and curious engagement with one's social and/or material surroundings and initiates individual and collective experiences of meaning, values and things other than "mine", and may thus facilitate a critical reception of one's own heritage [54] (p. 70). This is why writing it into policies or implementing it into practice requires reflection on the child's cultural identity. Is it already fixed or predetermined by the family's background or is it in the process of being made? Both the family and the ECE community can play an important role in answering.

### 3.2.2. Policy Analysis

As previously mentioned, the English, Welsh, Scottish, Irish and Swedish documents articulate the need for appreciation and celebration of children's belonging to diverse cultural communities outside the ECE settings. In such an understanding, each child seems to carry and represent a particular cultural difference, which, again, may develop clear expectations of the trajectory of identity. Even though the documents open up the category of collective identity as English or Welsh to diverse types of cultural belonging, these diverse types of belonging and identity seem to be assumed as fixed and seem to be presented and celebrated within the ECE context but not explored or negotiated. The Australian [25] and Norwegian [31] documents, through their exploration of diversity of heritage and ways of living and believing, do not associate a particular difference with a particular individual but treat diversity more as a social context, where *becoming* is happening through exploration of the existing diversity. This becoming is not expected to reproduce and preserve the home cultures of children but to allows the child to create their own sense of self at the intersection of diverse cultures, values and meanings. However, elsewhere in the Australian curriculum, it is stated that "children have a strong sense of identity" [25] (p. 26), which should not be compromised through their learning in ECE. In our opinion, this contradicts the explorative approaches, as these relate to diversity of cultures and the concept of becoming and facilitate a variety of ways of identity formation. This suggests that the Australian framework plan for ECE [25] generates two chains of meanings connected to diversity/difference. The first focuses on the preservation of cultures and the other focuses on exploring and facilitating the formation of diverse identities and becoming (where the latter is equivalent to the Norwegian framework plan [31]).

As stated in the Norwegian curriculum, "Staff shall explore and wonder at existential, ethical, religious, spiritual and philosophical questions together with the children" [31] (p. 55). This is intended to help "promote respect for human dignity by highlighting, valuing and promoting diversity and mutual respect. The children shall be able to discover that there are many ways in which to think, act and live" [31] (p. 9). These explorations and discoveries must, however, support the experience of togetherness and the value of community: "Kindergartens shall also give the children shared experiences and highlight the value of community" [31] (p. 9). The importance of everyone's participation is recognised. This focus on participation seems to be equivalent to the Slovenian [35] method of formulating diversity in the ECE context and providing every child with an equal opportunity for participation. This is slightly different from the Croatian [26] focus, which connects diversity with the children's competence in developing social and civic skills in accepting and understanding differences (arising from religious, racial, national, cultural and other differences or special needs). However, the Serbian curriculum [34] emphasises the importance of including minority cultures in institutional practices, and this may be seen as equivalent to the British [28,30,31,37] approach of including children's diverse cultural identities established outside the ECE settings; it is also similar to the Australian curriculum's belonging through the "context of the family" and "respect[ing] multiple cultural ways of knowing, seeing and living" [25] (p. 18).

*3.3. Local Places (and Communities)*

3.3.1. Theoretical Mapping

Sense of belonging, however, does not develop only among human beings but also between human and non-human elements, such as between humans and places. This is in line with the material perspective on social sustainability that is being advocated, according to which one should embrace not only people but also their inseparable exchange with their material, physical and natural surroundings [56]. These socio–material contexts may be seen as providing conditions and opportunities for social equity, as particular types of relationships between human beings and their environment can help to sustain a sense of connection, community and territoriality [57,58]. According to Rayner [59], space does not passively surround us. It is a vital, dynamic and complex element, allowing diverse possibilities for activity and communication, where both the people and the surroundings matter. This makes it possible to consider material elements and social relations as co-constituting each other. This suggests that the human subject cannot be seen as separate from the objects with which it is concerned [60] and intertwined and challenges any clear dichotomy between subject and object. The implication of this thinking for social sustainability (which focuses mainly on relationships between humans) is that it includes the non-human elements, even though these are systematically recognised as environmental and/or economic pillars of sustainability.

The lived human–non-human connection constitutes people's bond with and through place, whilst also enabling individuals to define and redefine themselves as they form communities in particular places, as well as across them. A sense of connection and attachment to place is, as argued by Pollmann [61], learned and habituated, yet open to modification and reconstruction through reflexive agency, educational practices and the acquisition of intercultural capital. This is in line with the description of sense of *belonging* as an "affective bond to particular geographic locations, and the meanings ascribed to such a bond changes over time, which develops a sense of belonging in people that makes a particular place an anchor of their identity" [62] (p. 3). Such experience of place is not only local; it is a source of meaning and affection.

Place can thus be understood as an arena for human everyday life and interaction [63], the shape and character of which "produces" the place [12]. Massey [12] describes places and landscape in terms of continuous change and dynamics and as essentially open and hybrid, always provisional and contested and transformed in line with people's activity and the (power) relations between them [9]. This will occasionally lead to a sense of loss [64] (p. 40) as well as to (a sense) of belonging. Some places, especially within educational institutions, may be "occupied" by particular gender and age groups [65], which, again, puts emphasis on potential mechanisms of segregation and exclusion. The public spaces of the local place, despite being public (or open to all), may be informally divided into places for "us" and "them". In such cases, a sense of place becomes an embodiment of the membership that underpins *belonging*.

The connection that children have with local place is formed through their participation in the local community's daily life, diverse structures and groups and in cultural arenas outside the ECE setting. This may provide an experience base for social learning and for common references and social equalisation [57,58]. Equitable access to community activities is crucial for social sustainability, connection to place, feelings of territoriality and belonging. Engagement with local surroundings can be linked to developing social and civic engagement [11,66,67]: for democratic consciousness to take shape, there must be something that concerns the individual, something the individual will take care of and develop into something better, to share with someone and make room for more people to participate [68]. Healthy and happy individuals with a strong sense of place, identity and hope for the future are more likely to make protection of their environment a priority [69].

3.3.2. Policy Analysis

Our analysis shows that the discourse on place and community that is present in all the documents does not mirror the theoretical complexity presented above. Rather, the material and natural surroundings are taken for granted in the analysed documents. However, comparing them allows reconstructions of different chains of meaning attached to a locality's importance.

The Norwegian curriculum distinguishes "*local community and society*" [31] as a learning area that should encourage active engagement with the ECE surroundings: "Through exploration, discoveries and experiences, kindergartens shall help the children familiarise themselves with their local community, society and the wider world" [31] (p. 36). Moreover, "Kindergartens shall give them knowledge and experience of local traditions, institutions and vocations so that the children feel they belong in their local community" [31] (p. 56). The Danish curriculum [27] seems to operate in the local community rather than in "the wider world", while the Swedish curriculum [36], again, seems to relate to learning about the wider world in terms of societal functions: "create conditions for children to become familiar with their surroundings and those societal functions that are important for every-day life and to take part in local cultural life" [36] (p. 16). Familiarising children with their local surroundings and institutions is also present in the Polish curriculum [32]; however, the curriculum seems to assume the urban character of the surroundings by referring to cultural institutions (such as theatres and museums) that are typical of urban spaces. Making the child familiar with them is seen as part of ECE's work in readying the child for school.

Familiarising children with their surroundings in the Croatian [26], Serbian [34], Slovenian [35] and Australian [25] curriculum is balanced with empowering children as active participants and agents who contribute to the local community. This may be seen as equivalent to the English statutory framework [28], in which children are active community-makers. They participate in and contribute to multiple communities as they move between home, extended family, ECE settings and play areas (p. 30).

The difference, however, lies in the assumed role of parents. In Poland [32], Croatia [26], Slovenia [35] and Serbia [34], parents are "a link" between the child, ECE and the local surroundings, and they play a crucial role in introducing children (both their own and others in these settings) to the locality. In the UK context, the children, themselves, are seen as the main actors as they move across and connect diverse communities and institutions with one another. "They often act as cultural brokers, helping families and settings understand one another" [29] (p. 24).

Despite the differences in the defining roles of the parents and children, the English statutory framework [28] directly articulates a meaning connected to places and spaces that seems to be tacit and assumed in the other policy documents: "Place, space, and histories are important. Communities and settings are embedded in particular places with their own geographies ( . . . ) Shared memories are often a source of comfort and solidarity, but they can also shadow the present by memories of injustice and hardship in the past" (p. 24). This is equivalent to the way that place is approached in the Australian curriculum [25], which points to the need to facilitate children's confident connection to familiar places and people and which is intended to further develop children's perseverance, resilience and optimism.

According to the Northern Irish "Curricular Guidance for Pre-School Education" [30], "children need an understanding of space in order to consider the relationships between objects" (p. 27), which is equivalent to the Norwegian [31] and Australian [25] understanding of place as a resource for learning to use natural and processed materials, which can also be seen as consistent with the focus on school readiness in the Polish curriculum [32]. It is, however, also very different to the understandings of place that emphasise the social and identity-related aspects of places.

Local place is not emphasised in the Croatian curricula [26], and the child is referred to as part of and a contributor to the community, in general. The Slovenian curriculum [35]

does not mention local place but does stress the importance of connection to socio–cultural and natural environments. The Serbian document [34] presents a broad list of local places (such as other educational institutions, health centres, cultural institutions and nearby craft centres) that children should be introduced to as a way of living within the environment.

### 3.4. Collaboration with Caregivers

#### 3.4.1. Theoretical Mapping

Various research-based recommendations have highlighted children's cognitive and non-cognitive outcomes, successful transitions into school and contributions to social inclusion as a result of parental involvement in ECE. All of these have been summarised in the systematic literature review by Moss, Lazzari, Vandenbroeck [70] and Bennett [38]. Bennet [38] additionally points out two pedagogical traditions within ECE: the preschool tradition and the social pedagogy tradition. The former involves parents in work and school readiness, while the latter sees the ECE setting as deeply contextualised within the local community. According to this understanding, parents are seen as the "bridge" between the ECE setting and the local community, supporting its way of functioning through diverse forms of collaboration, events and projects in and with the local community [15,16].

Elliot and Davis [71] acknowledge Bronfenbrenner's Ecological Systems Theory model [72] as groundbreaking, in terms of understanding human development within socio–political and cultural contexts. Its focus on the impact of human connections and relationships on the lives of children may also support a holistic pedagogical approach to children in collaboration with their families and further community. They also argue that interactions with physical or natural environments that shape children's experiences are mostly absent from Bronfenbrenner's model and that these systems need a deeper and broader interpretation of environmental needs. They propose new ways of representing/updating Bronfenbrenner's [71] work and present eco-pedagogical approaches that go beyond the anthropocentrism of Bronfenbrenner's theory. These also include different parental perspectives and a view of the broader local community as part of a community ecosystem in which the parts are interconnected [71].

Another body of knowledge addressing parental collaboration shows that institutions collaborate mostly and easily with local middle-class parents, which shows that there are cultural discourses involving the majority that underpin both the expectations and form of cooperation with caregivers [73–79]. Small qualitative studies have drawn conclusions that emphasise the importance of ECE practitioners fostering dialogue in which both parties provide explanations so as to understand one another's standpoint [75,80,81] and in which parents can offer support and individualised attention [80].

#### 3.4.2. Policy Analysis

ECE is obliged by the Norwegian curriculum to "work in partnership and agreement with the home to meet the children's need for care and play" [31] (p. 7). It is the responsibility of ECE to "facilitate co-operation and good dialogue with the parents" [31] (p. 29). In this dialogue, however, "both parents and staff must acknowledge the fact that the kindergarten has a social mandate and a set of core values and that it is the kindergarten's responsibility to uphold them" [31] (p. 29). Nevertheless, it is also the ECE setting that "must seek to prevent the child from experiencing conflicts of loyalty between home and kindergarten" [31] (p. 29).

This indicates that the home and the ECE setting are equal partners in the dialogue, as long as the parents agree with the core values of the document, which are democracy, diversity and mutual respect, gender equality, sustainable development, equality and equity [31]. This may make it sound as if these take precedence over other values potentially represented by the caregivers, as these are values that underlie the Western tradition of dialogue and democracy. However, they are made explicit so that it is transparent to all groups entering the ECE settings which value positions the institutional setting will represent and observe. The Slovenian curriculum [35], however, emphasises the importance

of showing a high level of respect for the values, languages and beliefs of all caregivers on their premises. The document does not, however, indicate precisely how possible parental values should be included in ECE content.

While the Norwegian [31], Swedish [36], Danish [27] and Australian [25] curricula point to reciprocal dialogue in partnership with parents in order to safeguard the holistic development of the child, the Australian curriculum regards families as "children's first and most influential teachers" [25] (p. 13), whereas the documents from Croatia [26], Poland [32] and Serbia [34] see the family as supporting ECE in the upbringing of children and the ECE settings as supporting the family in helping the children to learn. Evidently, in the Anglo–Celtic tradition, learning and school preparation are the object of greater parental involvement and parental cooperation, which safeguards the information exchange regarding the child's needs, the fulfilment of which is a condition for learning, as is the case in England and Australia [25]. Scotland [33] seems to go one step further by obligating ECE settings to provide support to the home in becoming a more learning-stimulating environment. As the Scottish document states, "supporting parents to provide a stimulating and supportive home environment, particularly in the early years, combined with high quality pre-school and school education is therefore a key element in delivering solidarity and cohesion and improving participation and productivity within the Scottish economy" [33] (p. 7).

All of the countries see collaboration with parents as supportive of children's learning and development, which is important for society, in general, but the ways in which this is organised differ. While Poland, Croatia, Serbia, Slovenia, Denmark and Norway address this collaboration through different opportunities for getting involved, such as exchanging information about the child or participating in and making decisions by way of parental boards (Poland, Croatia and Norway), the English statutory framework [28] and the Australian framework [25] indicate that parents are important cultural knowledge resources that inform the learning that takes place in the ECE setting "without compromising their [the children's] cultural identities" [25] (p. 26).

Here, again, comes the assumption that the children's and families' fixed identities make the family an expert in the child's cultural identity. This hegemony of meanings is not in the Norwegian framework plan [31], which sees the family as a resource for cultural knowledge but not as determining the identity of the child (which is in the process of becoming). The Serbian curriculum [34] develops this equivalence of meaning even further, stating that, as a result of a range of events in the recent history of the region, a single family is not capable of introducing the child to the complexity of cultural values lived and practised in the society, which is why ECE takes responsibility for this task.

## 4. Discussion

In this discussion section, we refer to the chains of meanings reconstructed in the policy documents and the theories mentioned at the beginning of each analytical section. In particular, we discuss how meanings occurring in one policy can visualise what is excluded in the other or how a set of meanings established in theory show what is excluded from policies and, as such, challenge them.

In relation to sense of belonging, the policy documents assume this to be either a processual or a fixed/fixable phenomenon, which guides ECE efforts to facilitate children "fitting in" or extending the criteria for experiencing membership in the group. However, the different hegemonies of meaning attached to belonging become equivalent in their assumption of a dichotomic character of belonging. Children are assumed to either belong or not. This "fails to capture the affirmed world of difference" [82] (p. 56), whereas a rhizomatic understanding of belonging [46] can embrace its different, ever-changing forms in a range of contextual aspects and circumstances. The ever-changing terrain of belonging may be influenced by a series of interconnected events or ways of living that make it possible to consider children's multiple belongings, their intensities and their human–non-human character.

The dynamics of multiple belongings bring us to the issue of identity raised in the analysis. The English [28], Welsh [37] and Australian [25] curricula assume that identity is home- and family-anchored and of a fixed and stable character and that it should not be compromised in the institutional setting of ECE. However, the Norwegian [31], Danish [27] and Serbian [34] documents indicate, as we understand them, the need for children to engage dialogically with diverse cultures and meanings so that they can explore, learn and *become* themselves. Sweden narrows the identity issue to language and includes this in the content of ECE without taking any position in relation to identity.

The identity-related assumptions in the analysed curricula invite one to reflect on the reservoir of identity-related meanings that have been excluded. In the case of the assumption of a fixed identity, the child's becoming is significantly limited and narrowed to learning that is locked inside the private sphere of family life. In countries where identity is seen as family-anchored, fixed and stable, such as England, Scotland, Wales and Australia, parents/caregivers are seen as experts in these issues. They are encouraged to offer their input in creating more inclusive environments to support their child's learning.

In the documents that do not assume that children's identities are determined by family background (such as the Norwegian, Danish, Swedish, Croatian, Serbian and Slovenian curricula), all contact by the children with diverse cultures and values is embraced, so that their identity formation, becoming and learning are facilitated. In these documents, children are not expected to preserve the culture of their families, as is the case in the English, Scottish, Welsh and Australian documents. The preservation of minority cultures does appear in the ECE content, however, as part of a diverse society.

The Polish curriculum does not mention cultural diversity at all, which, in a homogenous society, approximately 90% of which consists of Polish citizens, can be seen as silencing minorities and making practices of dealing with difference dependent on local contextual practices, implicit bias and the private (either prejudiced or affirmative) attitudes of professionals.

In the context of sense of belonging to place, some reconstructed chains of meaning have assumed place as something stable and fixed, as in Poland, Croatia, Serbia, Slovenia, Norway, Denmark and Australia, which children should be introduced to and made familiar with. The Nordic countries, in particular, have a long tradition of using nature and outdoor areas as a resource for work related to social competence, sustainable development and belonging [83].

Conversely, the perspective presented in the English [28] curriculum identifies children as agents and community makers because of their transitions between institutions and communities, and according to the Australian curriculum, the need for developing confidence occurs when entering diverse places where there is shared thinking and "collaborative learning" [25] (p. 18). This may be understood as a way of overcoming the exclusive character of particular places that are being occupied by particular groups of people or particular genders or positions. In Northern Ireland, the idea of place is directly connected to the non-human dimension of objects and materiality, as well as shared memories and sense of community, which encourages thinking of ECE settings in the local context as human and non-human assemblages [57,58] where learning and development take place.

## 5. Conclusions

From the normative standpoint of social sustainability, which emphasises the importance of equity and justice, it seems clear that policies that are oriented towards processual understandings of sense of belonging and pressure on ECE efforts to extend the criteria for child membership are more socially sustainable than others. Such policies may be strengthened by a more rhizomatic understanding of sense of belonging, which could potentially help practitioners to understand the heterogeneity of the diverse cultures in dialogue with which children become themselves, as well as the human–non-human (human–place) dynamics of which such heterogeneity is constituted. Understanding children's cultural identities as fixed and expecting them to be preserved by ECE or understanding them as in

development and expecting them to be supported by ECEs through access to different values, artefacts, ways of living and beliefs poses a dilemma in our view. Social sustainability aligns with cultural sustainability when cultural heritage is important; on the other hand, individual children should not be burdened with reproducing particular heritages. We thus see the concept of *becoming* as worthy of greater attention from policymakers, so that the child's self-creation as a subject with access to and in dialogue with diverse cultural values and meanings can be sustainable, as it also provides diversity as a joint reference (see also Section 3.2).

As parental identities become more stable and fixed, we view inclusion of their cultural knowledge in ECE content and practices as a matter of great importance. However, this must be done without prescribing a particular cultural heritage to a particular child but by using it as a resource for the whole group so that exploration, diverse identifications and formative development can take place. In relation to the concept of place, most of the documents emphasise its human, community-related character, while only the Northern Irish document [23] acknowledges the human–non-human assemblage. From a sustainability standpoint, connection with the non-human dimension of our world is important, and awareness of how the non-human aspect informs inter-human relationships is of importance and deserves a greater place in future policies on social sustainability through ECE.

These conclusions are limited since they are based on what social sustainability means for us today. Our intention was to demonstrate how social sustainability is indirectly addressed in ECE policy documents and how it is established through different hegemonies of meanings attached to sense of belonging, local place, diversity and difference, as well as through collaboration with parents and caregivers. By comparing these established sets of meanings, we hope to inspire the growth of new chains of meaning. This paper does not conclude by advancing one or another chain of meaning, but rather by advocating on behalf of the need for continuous comparative reflection, which enables diverse localities to function for one another as spaces for critical distance and thus unmask the excluded surplus of meaning and provide other perspectives and opportunities for the assessment of one's own policies. Therefore, we suggest approaching research as an arena for international dialogue on ECE policies, where not only can documents be compared but also policymakers, researchers and practitioners can have the opportunity to exchange meanings, co-create and inspire local policies.

Our recommendation for future socially sustainable writing of policy is to have international meetings/workshops that would allow policymakers to construct local policies on the basis of local ECE context, conditions and/or systems, while continuing to participate in global/international dialogue, as sustainability is of worldwide relevance. Such policy co-creation could be followed up with research, which is an alternative to today's dominant practice of research generating policy briefs, which policymakers create to a high degree and which limit the opportunities for authentic engagement with communicated meanings.

We view this as a fascinating area for further research, and we suggest following the structure of how diverse policies are implicated in institutional practice or, alternatively, how ECE settings work with social sustainability when they are not directly linking their own work to the value of social sustainability. Such studies could thus foster the creation of an overview of the ECE sector's impact on sustainable futures for diverse communities and show this sector as one that is particularly worthy of investment.

**Author Contributions:** Conceptualisation: A.R.S., Y.B., J.G. and A.V.-J.; Methodology: A.R.S.; Investigation: A.R.S., Y.B., J.G., A.V.-J. and J.K.: each of the authors investigated policy documents, which she had lingual and cultural access to. In case of documents without official English translations, the analysed parts were translated by one author to the rest of the group so that the whole group could participate in analysis. A.V.-J. translated to the author group the curricula from: Croatia, Serbia and Slovenia, and A.R.S. the Polish curriculum. After all documents were accessible by the whole group, the analytical work started. As the analyses were about joint reading and discussions, it is hard to distinguish clear contributions. Writing—original draft preparation: all of the authors contributed to

the original draft preparation, and it was A.R.S. who organised the final version of first draft. All of the authors were contributing to all of the sections, with particular responsibilities connected to particular policy documents: Y.B. was responsible for the Danish, Swedish and Norwegian Framework Plan; J.G. the ones from England, Northern Ireland, Scotland and Wales; J.K. the Australian Framework Plan for ECE; A.V.-J. for the Croatian, Serbian and Slovenian; and A.R.S. for the Polish and Norwegian. Writing—review and editing: A.R.S. led the conceptual process of revising the text in line with the reviews and coordinated everyone's contributions to the content. All of the authors contributed to the revisions. J.K. proofread and language edited the draft no. 2 (after major revisions), and A.V.-J. organised the references in the text and on the reference list. Project administration: A.R.S., with help of all authors. All authors have read and agreed to the published version of the manuscript.

**Funding:** Research Council of Norway, project no: 275575, Kindergarten Knowledge Centre for Systemic Research on Diversity and Sustainable Futures (2018–2023).

**Institutional Review Board Statement:** Not applicable.

**Informed Consent Statement:** Not applicable.

**Data Availability Statement:** The data are the policy documents available on each country's government pages.

**Acknowledgments:** Kindergarten Knowledge Centre for Systemic Research on Diversity and Sustainable Futures (2018–2023); EECERA Special Interest Group: Children from Refugee or Migrant Backgrounds.

**Conflicts of Interest:** The authors declare no conflict of interest.

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
