# Peer review of "Unfreezing the Discursive Hegemonies Underpinning Current Versions of “Social Sustainability” in ECE Policies in Anglo–Celtic, Nordic and Continental Contexts"

_sustainability, doi:10.3390/su13094758_

Round 1
Reviewer 1 Report
Dear authors, the manuscript "Unfreezing discursive hegemonies underpinning current versions of “social sustainability” in ECEC policies in Anglo-Celtic, Nordic and Continental contexts" is a compelling read and an important contribution to the developing/evolving field of early childhood education and care and its relation to social sustainability. The strength of the manuscript is its focused and clear comparative analysis of commonly used ECEC policy discourses related to social sustainability as it is understood and taken up in the context of early education (e.g., "sense of belonging"). By providing and juxtaposing examples of the manifestation of ECEC policy discursive practices related to social sustainability from eight countries, the reader is invited to critically reflect and engage with the text, make inferences, and experience the power of discourses to produce (different) realities.
My main suggestion is to elaborate on and theoretically contextualize a number of concepts that are mentioned too briefly and without their theoretical grounding. Those include the notions of becoming, assemblage, and mattering (note that assemblage and mattering appear as keywords even though they are hardly mentioned in the text). After elaborating on the aforementioned concepts in the text, the conclusion can be strengthened by returning to these concepts and explaining their relevance to the "Unfreezing of discursive hegemonies."
In the abstract you say: "We conclude with an argument that continuous reflection and reflection and reflexivity is more important for social sustainability..." and connect this statement specifically to researchers. Might it be relevant to also tie this to educators who are the ones "operationalizing" the reviewed policies?
Finally, acronym use was inconsistent throughout in terms of using ECE and ECEC. Choose one and include all words when it is first used before using an acronym.
"This paper does not conclude by promoting one or another chain of meaning"
Author Response
Dear Reviewer,
We are very thankful for your constrictive feedback, which in combination with other reviews allowed us to reflect on and improve the text. The major revisions we did in the text are about:
- More solid theoretical elaboration on the connection between social sustainability and sense of belonging (as well as the way we extend the sense of belonging with, as: diversity, local places and communities, as well as collaboration with parents/caregivers). This elaboration is now supported with a ray of references (and not just stated as before).
- Making it more clear that the paper is a policy analysis, which is why our conclusions can not refer to how policies shall be implemented in practice. However, we suggest ideas regarding the practice of policy making, which we find more sustainable. (this improvement is a response to comment about practical implication of our research as well as question about future research questions rising from our study).
- As the amount of text about different countries seemed unbalanced (however was related to the fact how much each of the document says about one or another category), we provide the reader with a table with an overview over our interpretation of the documents’ understanding of the analyzed concepts.
- We corrected refence style as well as unified the ECE and ECEC acronyms.
With kid regards,
Alicja R. Sadownik
Yvonne Bakken
Josephine Gabi
Adrijana Višnjić-Jevtić
Jennifer Koutoulas
Reviewer 2 Report
Thank you for the opportunity to study this research.
Overall, this is an interesting topic that is currently an important part of Early Childhood Education.
I have the following comments on the article:
- abstract - first abbreviation - Early Childhood Education - in my opinion ECE should be instead of ECC.
- I recommend expanding future research into the discussion section. What other questions caused your results and how to solve them.
- I also recommend creating a separate section of practical implication, which would raise the article one level up.
Section: Author Contributions, Funding, Data Availability, Acknowledgments and Conflicts of Interest are empty.
Author Response
Dear Reviewer,
We are very thankful for your constructive feedback, which in combination with other reviews allowed us to reflect on and improve the text. The major revisions we did in the text are about:
- More solid theoretical elaboration on the connection between social sustainability and sense of belonging (as well as the way we extend the sense of belonging with, as: diversity, local places and communities, as well as collaboration with parents/caregivers). This elaboration is now supported with a ray of references (and not just stated as before).
- Making it more clear that the paper is a policy analysis, which is why our conclusions can not refer to how policies shall be implemented in practice. However, we suggest ideas regarding the practice of policy making, which we find more sustainable. (this improvement is a response to comment about practical implication of our research as well as question about future research questions rising from our study).
- As the amount of text about different countries seemed unbalanced (however was related to the fact how much each of the document says about one or another category), we provide the reader with a table with an overview over our interpretation of the documents’ understanding of the analyzed concepts.
- We corrected refence style as well as unified the ECE and ECEC acronyms.
With kid regards,
Alicja R. Sadownik
Yvonne Bakken
Josephine Gabi
Adrijana Višnjić-Jevtić
Jennifer Koutoulas
Reviewer 3 Report
I am happy to read this manuscript. The topic, social sustainability, is interesting and important.
The manuscript seems to need some more polishing and rewriting. Here are some examples how to continue with this manuscript.
On the p. 2 writers gives even twice the list of operationalisational concepts. One of those lists is not needed. There are only seven rows time to forget them. However, there is very narrow and loose argumentation to connect given operationalisation to social sustainability. The frame, links between social sustainability and sence of belonging, local places, cultural diversity and cooperation with children’s homes (in abstract) or caregivers (in introduction) need to be more solid.
There are no explicit formulation of research question(s) in the beginning of manuscript. Reader can think and make logic conclusions - but it will help reader, if you really write Your research questions clearly in the beginning of the text. Without research questions it is rather difficult to find a clear focus in analysis 3.1-3.4. The argumentation of research question is not yet clear and strong enough. What is actually the point of this comparative research? I think writers can find a way to write it. Now writers actually are telling one form of the research question in the end of this text.
The methodology has to be improved. There are good methodology and literature about how to analyze documents. The description of analysis is now narrow and there are no reasoning nor argumentation why to follow described procedure.
Writers tell (row 69) that they are analyzing documents of eight countries: Australia, Croatia, Denmark, Norway, Poland, Serbia, Slovenia, Sweden and UK. I think there are nine countries on the list. How did you chose those countries and documents? I think writers can describe the process of choosing those countries more clearly. The analysis of so many documents seems to be challenging. After reading the third chapter, I ask myself, is it about countries or is it about documents? If the article is about analyzing documents, writers have to focus on documents.
And one more question: Is it relevant and acceptable to write about Northern Ireland by using the word “Irish”?
In 3.1 and 3.2 writers has different way to use references. Some sentences need references in 3.2.
The analysis in yet not in balance. There is quite a lot of information about Norway and information from Serbia, Croatia and Slovenia is very narrow. Reader needs more balanced citations and contents from each document. All documents need to discuss equally. Maybe a table could help to make it visible?
The idea of this paper, to compare several policy documents by four concepts, is challenging and ambitious. After rewriting this paper will be more informative, better argumented and several readers can learn of it. I hope writers can continue this process. I am looking forward to seeing next version. All the best for the process.
Author Response

(The authors gave the same response as above.)

Round 2
Reviewer 3 Report
Writers have done several improvements in this manuscript and it is clearly developed version. Table 1 gives now a good and informative base for reader of all countries. Some added references reinforce the quality of theoretical background. There are some comments and minor notes to consider.
Some additions, (such as 509-512) help reader to follow analysis.
Table 1 helps reader. However, the ambitious aim to compare so complex phenomena by so many countries is just difficult to handle without any problems - so the writing is partly challenging for a reader. The content is there, but diverse lists of countries are difficult to remember concerning the whole dataset. So it is relatively difficult to build a picture in ones mind and to understand various similarities and differences between so many countries. Thus, this text is now better and it is just one problem for clarity and it comes from design. Sometimes it happens and in this case the reader has to accept those solutions made by researchers.
Still there is the line 98. Writers say: "In order to answer this articles question referring to social sustainability in ECE curricula... ". I can not see a question. If you are asking something, could You, please write really the question. Or is it about something else? In order to compare...? In order to describe...?
Typo: r. 132 too many numbers in reference mark
I ask you to be consistent with reference list. If a journal wants doi-numbers, please use doi-numbers always, when you can find it. (Reference 40 - provided, 83 not provided - but easy to find: DOI: 10.5617/nordina.6186. )
Despite those problems of comparative design with so many countries i hope you can manage this process.
Author Response
Dear Reviewers and Editors,
we are very thankful for one more possibility for improving the text. The attached table presents overview over your reviews and conducted revisions.
With kind regards
(on behalf of the author group)
Alicja R. Sadownik
